# Suitability of Airborne and Terrestrial Laser Scanning for Mapping Tree Crop Structural Metrics for Improved Orchard Management

**Dan Wu [1], Kasper Johansen [1,2,*], Stuart Phinn [1] and Andrew Robson [3]**

[1]  Remote Sensing Research Centre, School of Earth and Environmental Sciences, The University of Queensland, St Lucia, QLD 4072, Australia; d.wu@uqconnect.edu.au (D.W.); s.phinn@uq.edu.au (S.P.)

[2]  Hydrology, Agriculture and Land Observation Group, Water Desalination and Reuse Center, King Abdullah University of Science and Technology, Thuwal 23955-6900, Saudi Arabia

[3]  Applied Agricultural Remote Sensing Centre, School of Science and Technology, University of New England, Armidale, NSW 2351, Australia; arobson7@une.edu.au

*  Correspondence: kasper.johansen@kaust.edu.sa; Tel.: +966-545351582

**Abstract:** Airborne Laser Scanning (ALS) and Terrestrial Laser Scanning (TLS) systems are useful tools for deriving horticultural tree structure estimates. However, there are limited studies to guide growers and agronomists on different applications of the two technologies for horticultural tree crops, despite the importance of measuring tree structure for pruning practices, yield forecasting, tree condition assessment, irrigation and fertilization optimization. Here, we evaluated ALS data against near coincident TLS data in avocado, macadamia and mango orchards to demonstrate and assess their accuracies and potential application for mapping crown area, fractional cover, maximum crown height, and crown volume. ALS and TLS measurements were similar for crown area, fractional cover and maximum crown height (coefficient of determination ($R^2$) $\geq$ 0.94, relative root mean square error (rRMSE) $\leq$ 4.47%). Due to the limited ability of ALS data to measure lower branches and within crown structure, crown volume estimates from ALS and TLS data were less correlated ($R^2$ = 0.81, rRMSE = 42.66%) with the ALS data found to consistently underestimate crown volume. To illustrate the effects of different spatial resolution, capacity and coverage of ALS and TLS data, we also calculated leaf area, leaf area density and vertical leaf area profile from the TLS data, while canopy height, tree row dimensions and tree counts) at the orchard level were calculated from ALS data. Our results showed that ALS data have the ability to accurately measure horticultural crown structural parameters, which mainly rely on top of crown information, and measurements of hedgerow width, length and tree counts at the orchard scale is also achievable. While the use of TLS data to map crown structure can only cover a limited number of trees, the assessment of all crown strata is achievable, allowing measurements of crown volume, leaf area density and vertical leaf area profile to be derived for individual trees. This study provides information for growers and horticultural industries on the capacities and achievable mapping accuracies of standard ALS data for calculating crown structural attributes of horticultural tree crops.

**Keywords:** airborne laser scanning; terrestrial laser scanning; horticulture; tree crops; crown structure; crown area; fractional cover; tree height; crown volume

## 1. Introduction

The canopy structure of a forest and individual trees influence ecosystem function, carbon cycle, biophysical processes, wildlife habitat, tree health and productivity [1–3]. For horticultural tree crops, canopy structure is also important, as it relates to yield, tree condition, light interception, pruning

requirements, irrigation and fertilizer application, and orchard management practices [4–8]. Differing from forests, horticultural tree crops generally have very short tree stems that have been mechanically altered via limb removal, and as such, most of the biomass is concentrated in the canopy, and therefore, canopy structure metrics are of significant interest to growers [9,10]. While forest inventory tends to focus on crown detection, height and stem diameter, crown characteristics including crown height, diameter, area and shape are desired parameters for orchard inventories [11]. For horticultural tree crops, canopy structure influences light interception and distribution within tree crowns, which impact yield, fruit quality and in some cases, pest and disease susceptibility [5]. The geometric character of orchard trees is also important, as it can guide pruning, irrigation, fertilization and pesticide applications and indicate tree health condition and vegetative growth [4,10,12]. It is complicated and time-consuming to measure crown structure manually [13]. Although remote sensing technologies have been applied to horticultural tree crop environments over the last decades, the main focus of remote sensing technologies on mapping canopy structures is in forestry [4,14].

Light detection and ranging (LiDAR) data have been recognized as the most precise and reliable technology for canopy structure mapping, as it provides both horizontal and vertical structure information at the forest and individual tree level [5,12,15–17]. Although LiDAR data have been applied for structural mapping of horticultural tree crops [4], there is limited information on the level of detail that LiDAR data can provide and how it may be applied to tree crops in a precision agricultural setting. While airborne laser scanning (ALS) data have been used for mapping tree crop structure, including crown height, crown dimension and crown volume [11,18,19], the majority of horticultural applications have focused on olive and apple trees [9,11,18,19]. Estornell et al. [9] demonstrated that low point density ALS data ($\sim$0.5 points/m$^2$) have the potential to estimate wood volume (coefficient of determination ($R^2$) = 0.70) and crown height ($R^2$ = 0.67) of olive trees. With an average point density of 4 points/m$^2$, Hadas et al. [11] estimated crown height (average error = 19%), crown base height (average error = 53%), length of the longer diameter and perpendicular diameter (average error = 13% and 9% respectively) of 25 olive trees. Estornell et al. [18] suggested that medium point density (4 points/m$^2$) ALS data underestimated crown volume due to underestimation of maximum crown height and overestimation of the lowest part of the tree crowns, while crown area was accurately mapped. Crown area and the maximum ALS intensity values within each tree were identified as the most important parameters to predict the pruning residual biomass when compared against field measurements ($R^2$ = 0.89, Root-mean-square error (RMSE) = 2.78 kg) [18]. Jang et al. [19] illustrated that ALS data underestimated apple tree crown height by about 1 m when compared to field measurements, but the ALS data (horizontal point spacing < 50 cm) successfully detected 99.4% of trees, except for those with a height below 1 m. Jang et al. [19] also acknowledged that factors such as crown height, irregular crown form, centre-opened crown shape and overlapping branches affected crown detection accuracy.

Terrestrial laser scanning (TLS) data have been used for olive and walnut trees to map individual crown architecture and advance orchard management [20,21]. Moorthy et al. [21] selected 24 olive trees to demonstrate that TLS data can produce robust and highly accurate individual tree crown architecture information. The crown architecture information included tree height, i.e., the difference in laser pulse reflection from the top of the crown and the ground ($R^2$ = 0.97, RMSE = 0.21 m), crown width ($R^2$ = 0.97, RMSE = 0.13 m), crown height (the top and bottom extents of the crown) ($R^2$ = 0.86, RMSE = 0.14 m), crown volume ($R^2$ = 0.99, RMSE = 2.6 m$^3$) and Plant Area Index (PAI) ($R^2$ = 0.76, RMSE = 0.26), defined as the total one-sided leaf and woody area [22]. They suggested that the TLS system should replace traditional manual field measurements to generate crown structure information for horticultural tree crops. Fernández-Sarría et al. [20] demonstrated a high correlation between field and TLS calculated olive tree crown structure information such as crown height ($R^2$ = 0.85), crown diameter ($R^2$ = 0.92) and crown volume ($R^2$ = 0.87). Similar results for these crown structure parameters were found in studies of walnut trees, therefore affirming the ability of TLS data to derive accurate crown structure parameters in horticultural tree crops [23]. Estornell et al. [23] calculated crown volume, crown diameter and crown height with high accuracy for walnut trees using TLS

data. These structure measurements from TLS data showed strong correlations to field measurements. They also indicated that further research needs to be conducted regarding the most suitable point density of TLS data required to extract different crown architecture information of horticultural tree crops, because the size of the TLS dataset can significantly influence the data collection time, the storage costs and efficiency of data processing. Wu et al. [6] derived tree level leave area (LA), leaf area density (LAD) and vertical leaf area profile of mango, avocado and macadamia tree crops. This study demonstrated that TLS technology has the ability to quantify LA, LAD and vertical leaf area profiles for horticultural tree crops and the LA change derived from TLS data were consistent with the expected LA changes caused by canopy management, growth and a severe storm.

ALS and TLS data have been compared for their ability to derive canopy structure information in forest plantations [24,25]. Factors such as point density, laser footprint size, scan angle and pulse power can limit the ability of ALS data to measure lower parts of the canopy and stem structure [24]. However, ALS is better suited for assessing the upper canopy due to the aerial perspective during data acquisition, while TLS can provide more detailed information about the lower canopy with potential occlusion of the tree apex, depending on tree density, structure and scan angles [25]. Korhonen et al. [26] estimated crown volume of 77 trees (mainly Scots pine, Norway spruce and birches) in a boreal forest in southern Finland from ALS data. Due to insufficient returns from the lower canopy, significant underestimation of crown volume (−24.7% on average) was found when compared against field measurements. TLS data can provide a much higher spatial resolution than ALS data but with limited spatial coverage, while ALS data can cover a large area but with a lower point density than TLS data [25].

A thorough review of current literature failed to identify any publications comparing ALS and TLS data for horticultural applications and estimation of crown structure of tree crops. In this paper, we address this knowledge gap by calculating crown structure information for avocado, macadamia and mango trees using both ALS and TLS data, including crown area, fractional cover, maximum crown height, and crown volume. In addition, due to different data resolutions and spatial coverage, LAD, LA and vertical leaf area profiles were calculated from the TLS data only, while tree row dimensions, crown height and number of trees at the orchard level were calculated from the ALS data only. The main objectives of this paper were to: (1) map tree crop (mango, avocado and macadamia) structure information, including crown area, fractional cover, crown height and crown volume, from ALS data and evaluate these measurements against TLS data; and (2) compare the practicality, vertical and horizontal coverage, and scalability of ALS and TLS data for calculating tree crop structure information for improved orchard management. Based on those objectives, we hypothesize that ALS data can be used to derive orchard relevant information on crown area, fractional cover, height and crown volume when evaluated against measurements obtained from TLS data, and that ALS data have capabilities for deriving tree row and orchard scale information such as row dimensions and number of trees. This research provides novel findings on mapping tree structure metrics from ALS and TLS data that may guide growers and agronomists on horticultural tree crop applications.

## 2. Study Areas and Datasets

### 2.1. Study Area

The Bundaberg region is one of the largest horticultural regions in Australia, producing a large variety of fruit and nuts from tree crops [27]. It has a subtropical climate, with a mean annual rainfall of 1022 mm (1942–2019), mean maximum temperature of 26.80 °C (1959–2019) and the mean minimum temperature of 16.40 °C (1959–2019) [28]. Bundaberg produces about 4000 tons of mangoes a year [29]. It is also one of three major avocado production regions, and the largest and fastest macadamia-growing region in Australia [30–32]. For this study, two avocado (*Persea americana*) cv. Hass trees from commercial orchards and two avocado (*Persea americana*) cv. Hass trees from a research station, one macadamia tree (*Macadamia integrifolia*) cv. Hawaiian Agricultural Experiment Station

(HAES) 344, and two mango trees (*Mangifera indica*) cv. Calypso, including a high and low vigour tree, were selected as representative samples for TLS data collection within the Bundaberg region (Figure 1). ALS data were collected for the orchards within which these selected trees occurred. The mango, avocado and macadamia orchards covered 1.22, 28.09 and 13.54 ha, respectively. The mango and macadamia orchards had a spacing of 4 m between trees within a row, while the avocado trees were planted 5 m apart.

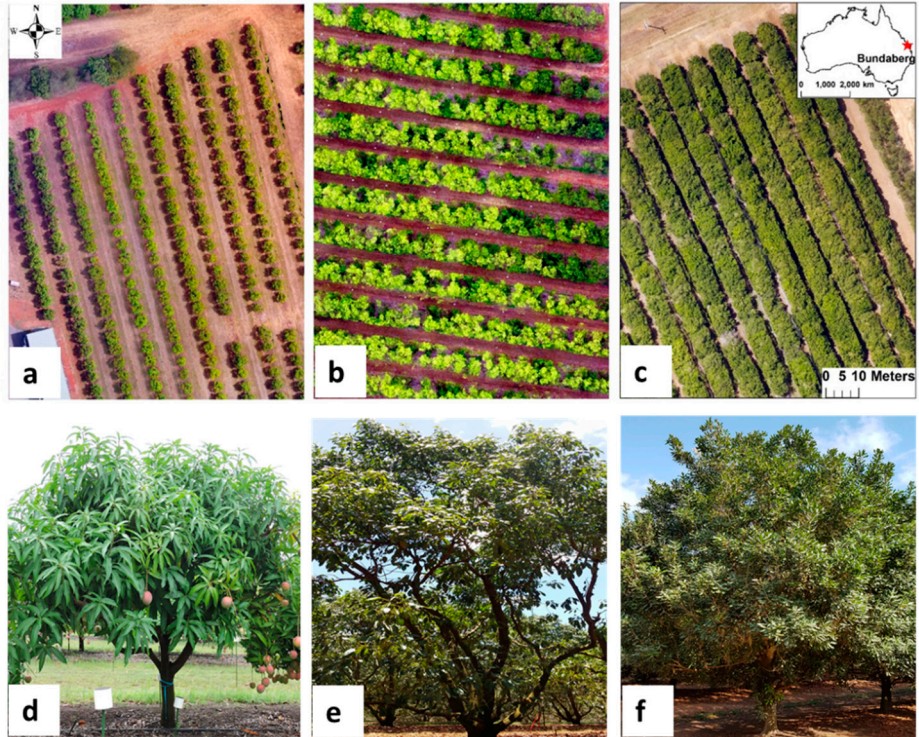

**Figure 1.** Unmanned aerial vehicle (UAV) imagery of (**a**) mango, (**b**) avocado and (**c**) macadamia orchards, and associated field photos of (**d**) a mango, (**e**) avocado and (**f**) macadamia tree.

*2.2. Datasets and Methods*

2.2.1. Datasets

Crown structure information on avocado, macadamia and mango tree crops was collected from a RIEGL VZ-400 (RIEGL Laser Measurement Systems GmbH, Horn, Austria) TLS system on 14 and 15 August 2016. The RIEGL VZ-400 TLS system was mounted on a tripod at a height of approximately 1.5 m. With a laser wavelength of 1550 nm in the near infrared part of the spectrum and a beam divergence of 0.35 mrad, the RIEGL VZ-400 scanner sends out laser pulses that can reach up to a distance of 350 m and records up to four returns per emitted pulse. Through inclination sensors and an internal compass, the RIEGL VZ-400 also collects pitch, roll and yaw information. The RIEGL VZ-400 settings are provided in Table 1 and the accuracy of the scanner is ±5 mm. The scan resolution of the TLS data is 0.06°. Four scan locations were set up around each tree to minimize occlusion. Due to the narrow row spacing and the scanner's zenith view angle (30–130°), a vertical and a 90° tilt scan were conducted at each scan location for the two mature avocado trees from the commercial orchard and the macadamia tree to ensure that the entire tree was scanned. Due to the time required to collect high-resolution TLS data from multiple scan angles (to minimize occlusion), seven trees were selected for evaluating the tree structural parameters derived from the ALS data. Reflector targets were set up around each tree and were visible from all scan locations. These targets were used to register and merge the TLS data collected at all four scan positions for each tree. Further details about the fieldwork procedures can be found in [6].

**Table 1.** RIEGL VZ-400 TLS scanner vs. RIEGL LMS-Q 1560 ALS scanner settings and data acquisition parameters.

| | RIEGL VZ-400 TLS Scanner | RIEGL LMS-Q 1560 ALS Scanner |
|---|---|---|
| Beam divergence | 0.35 mrad | <0.25 mrad |
| Pulse repetition rate | 300 kHz | 400 kHz |
| Laser wavelength | 1550 nm | 1064 nm |
| Minimum range | 1.5 m | 50 m |
| Maximum range | 160 m (at 20% target reflectance) 350 m (at 90% target reflectance) | 3500 m (at 20% target reflectance) 5100 m (at 60% target reflectance) |
| Field of view | 0°–360° (azimuth range) 30°–130° (zenith range) | 58° |
| Recorded data | Full waveform & up to four returns per emitted pulse | Full waveform & up to seven returns per emitted pulse |
| Accuracy | ±5 mm | ±20 mm |

The ALS data were acquired on 31 July, 2016 using an airborne small-footprint RIEGL LMS-Q 1560 LiDAR system with the laser scanner collecting data at 1064 nm (Table 1). The average flying height was 600 m above ground level with a pulse repetition of 400 kHz, an off-nadir angle of 30 degrees and a beam divergence of 0.5 mrad. These acquisition settings yielded a point density of 13.63 points/m$^2$. The vertical and horizontal accuracies were determined to be 0.029 m and 0.018 m, respectively, based on a calculation by the data provider of 120 ground control points.

2.2.2. Terrestrial Laser Scanning Data Processing

The TLS data were registered to the ALS data using the RiSCAN PRO (RIEGL, Horn, Austria) coarse registration and the multi Station Adjustment tools. The "all nearest points" mode was selected, and registration parameters, including search window radius, point cloud rotation angle, minimum and maximum adjustment errors, outlier threshold and the least square fitting calculation mode were used to perform the multi Station Adjustment. Registration errors were between 0.03–0.09 m for the seven assessed trees, which were considered negligible in relation to the tree crown dimensions. Due to the overlapping and continuous hedgerow canopy, a bounding box was created for each avocado and macadamia tree to make sure that the ALS and TLS data were clipped to the same extent. The TLS point cloud was classified into leaves and branches based on their geometrical properties, using the CANUPO segmentation algorithm provided by the CloudCompare™ software (version 2.9.1, General Public License software, http://www.cloudcompare.org/). The classification parameters are provided by [6], which yielded accuracies of 94% for the macadamia tree and 99% for the mango and avocado trees. Subsequently, manual corrections were undertaken, as some points were incorrectly classified as leaves around first branches of the tree crops. Classified point clouds were then used to calculate the vertical leaf area profiles as well as LA and LAD at the voxel (a three-dimensional (3D) equivalent of a pixel) level. Here, LA, which was defined as the one-sided total leaf surface area, was calculated using the formula LAD*(voxel side length)$^3$ [6,33,34]. A voxel side length of 25 cm was chosen based on the tree crown sizes to demonstrate the voxel level LA and LAD calculations based on crown geometry and leaf sizes. Details of the data registration, classification, LA, LAD and vertical leaf area profile calculation methods can be found in [6,35].

To calculate crown area, the crown point clouds were triangulated in the LAStools software (rapidlasso GmbH, Gilching, Germany), i.e., las2tin along the convex hull of each of the point clouds. Then these triangular irregular networks (TIN) triangles were merged into a single polygon for each crown to calculate its crown area. TINs are vector-based data constructed by triangulating a group of points [36]. To calculate fractional cover from the TLS data of the vertical view to relate to the ALS data, we firstly calculated a canopy height model (CHM) at 0.1 m spatial resolution based on the point density. Assuming that there was no occlusion, based on the crown size we gridded the CHM into 0.5 m grids to calculate the percentage of crown pixels that had a value higher than 0.15 m within each

grid to derive the fractional cover. The crown height threshold was set to 0.15 m to omit understory and ground features in the orchards. TLS points were classified as ground points and non-ground points first using "lasground" and maximum crown height was calculated using "lasgrid" within the LAStools software, where height was determined by the highest point above the ground TIN at their x and y location. The crown point clouds were also used to calculate crown volume. Voxels with a side length of 25 cm, deemed suitable for the crown size of the tree crops, were created based on point clouds belonging to each crown, and then the crown volume was calculated based on counting the filled voxels using the LAStools software. A detailed flowchart of the TLS data processing steps for measuring crown structure metrics of the individual tree crowns is presented in Figure 2.

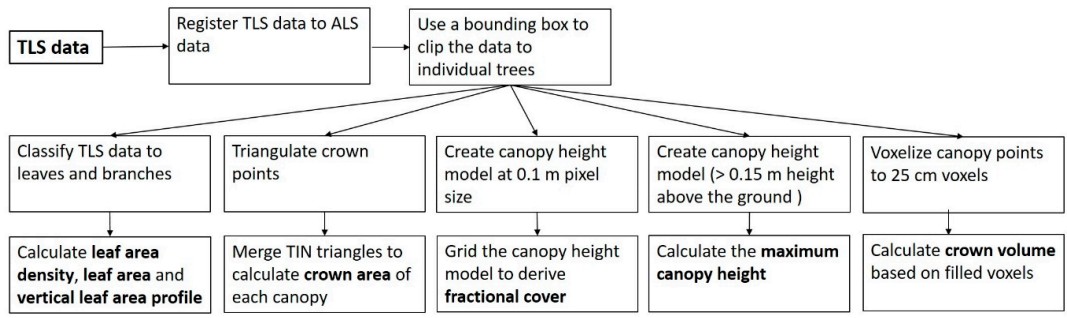

**Figure 2.** Flowchart for generating crown structure metrics from the terrestrial laser scanning (TLS) data.

### 2.2.3. Airborne Laser Scanning Data Processing

At the orchard level, a maximum CHM was created for each avocado, macadamia and mango orchard. By selecting 0.15 m as the canopy height threshold, the CHM was converted into a shapefile representing the hedgerows, and manual editing was conducted to separate the connected rows due to branches reaching across tree rows. A minimum-bounding rectangle was created for each tree row in ArcGIS 10.6. The tree row width and length were represented by the dimensions of the rectangle. From the length of the rectangles and knowing the spacing between tree crowns from field assessment, the number of trees per tree row and orchard were calculated. At the tree level, individual trees were extracted from the ALS data using the same bounding boxes as those used to clip the TLS data. Fractional cover was derived from the proportion of first returns (higher than 0.5 m) in relation to all returns [37]. All ALS points except for the ground points of each clipped individual tree was used to calculate crown volume using the same voxel-based approach as for the TLS data. The voxel-based approach was used in this case, as it is a direct measurement of points as opposed to model-based estimates. The same workflow was used for both the ALS and TLS for deriving crown area, maximum crown height and crown volume (Figure 2).

### 2.2.4. Evaluation of ALS-Derived Tree Crop Structure

In this research, TLS-derived measurements of tree structure were used for evaluation of the ALS-derived results. The TLS data were deemed suitable for evaluation purposes based on existing research findings, e.g., by [38,39], and the much higher point cloud density, smaller laser footprint size, integration of multiple scans of each tree from different view angles and measurement proximity to each tree. We used linear regression to relate the ALS-derived results of the mapped crown structure parameters to those produced from the TLS data of the corresponding trees. We used the $R^2$ and the line slope and intercept for interpretation of the relationship. A two-tailed t-test was performed to evaluate if the intercept and slope of the equations of the line of best fit was significantly different from zero and one, respectively, at a significance level of 0.05 [40]. Also, RMSE was calculated to indicate the spread of the residuals, expressed as the standard deviation between estimated crown structure values derived from the ALS and TLS data, using a linear model. In order to compare the linear models between different crown structure measurements calculated in different units (i.e.,

crown area, m$^2$; crown height, m; and crown volume, m$^3$), we also calculated the unitless relative RMSE (rRMSE), by dividing the RMSE with the respective mean crown structure value estimated from the TLS data. Hence, the lower the rRMSE is, the better model fit. For crown area evaluation, the percentage difference of the ALS and TLS measurements were calculated. For tree crown fractional cover and height, the values of individual pixels encompassed by each tree crown's perimeter were related to identify the maximum and minimum differences, average values as well as the RMSE for each individual tree. The number of trees estimated per tree row and orchard from the ALS data were evaluated against field counted numbers of trees for selected tree rows, including the entire mango orchard, 10 avocado tree rows, and 5 macadamia tree rows.

## 3. Results

Our results showed that those crown structure measurements relying on information from the top of the crown were largely similar when using both the ALS and TLS data (Figure 3). The measurements of crown area achieved the highest correlation (R$^2$ = 0.997) between the ALS and TLS data (Figure 3a). The RMSE of the crown area derived from the ALS data evaluated against the TLS data was 1.11 m$^2$ and the rRMSE was only 4.47%. Low estimation differences between the ALS and TLS-derived fractional cover (rRMSE = 3.46%) were also observed (Figure 3b). The RMSE of fractional cover determined from the ALS data was only 0.03 when assessed against TLS measurements and the R$^2$ was 0.94. The RMSE of the linear model between maximum crown height derived from the ALS and TLS data was 0.29 m and the R$^2$ was 0.99 (Figure 3c). A linear model for maximum crown height estimates between the ALS and TLS data achieved the best model fit (rRMSE = 2.59%) of all the linear models of crown structure with a near 1:1 relationship. In fact, the relationships between the ALS and TLS estimates of crown area, fractional cover and crown height all had near 1:1 relationships intercepting close to the origin. Based on a two-tailed t-test, it was found that the slope was not significantly different from 1 at a significance level of 0.05 for crown area, fractional cover and maximum height. The intercept was not significantly different from 0 for fractional cover, but the null hypothesis of the intercept was rejected for crown area and maximum height at a significance level of 0.05. On the other hand, crown volume measured from the ALS data was consistently smaller than that derived from the TLS data for all tree types (Figure 3d), with the slope and intercept being significantly different from 1 and 0, respectively, at a significance level of 0.05. While the R$^2$ was 0.81 when evaluating the ALS against the TLS-derived crown volume, the linear model had the poorest fit (rRMSE = 42.66%) of the four crown structural parameters. In fact, the ALS-derived crown volume estimates were >10 times smaller than those derived from the TLS data, which was attributed to the top-down viewing geometry of the ALS data, preventing volume estimation of the lower parts of the tree crowns (Figure 3d).

### 3.1. Evaluation of ALS Crown Area Against TLS Data

The crown area of the avocado, macadamia and mango trees was derived in the same manner for both the ALS and TLS data, using a TIN merged into a single polygon from which the crown area was calculated. Crown areas estimated from the ALS data were consistently smaller than the crown areas derived from the TLS data for all avocado, macadamia and mango trees in this study (Figures 3a and 4). The maximum absolute crown area difference between the ALS and TLS data was 4.16 m$^2$ (9.15% difference) for one of the avocado trees in a commercial orchard (Figure 4a), while the minimum absolute crown area difference was 0.43 m$^2$ (7.29% difference) for the low vigour mango tree (Figure 4d). The largest and smallest percentage differences between the ALS- and TLS-derived crown area was 22.00% for the high vigour mango tree and 2.67% for the macadamia tree, respectively. On average, the ALS crown area was 1.90 m$^2$ smaller than the TLS crown area for all the trees in this study. Estornell et al. [18] showed that due to the lower point density, ALS data may not be able to detect the border and lower part of the crown when mapping the individual tree crown area for horticultural tree crops. Similar findings can be seen from our study for all the three tree crop types and the significant difference in point density between ALS and TLS data is illustrated in Figure 4.

Furthermore, compared to the overhead vertical data acquisition method of the ALS data, the side view data acquisition of TLS data was better suited for capturing the border and lower parts of the tree crowns, especially for vertically overlapping crown layers. For example, ALS data could not be used to detect the southern edges of the crown for the high vigour mango tree, which was occluded by the crown from the adjacent tree (Figure 4d). Therefore, the lower point density and vertical view angle contributed to the limited ability of ALS data to detect the edges along the crown parameters, causing the ALS data to underestimate the crown area for all the horticultural tree crops. On the contrary, ALS data could be used to accurately identify the crown area of the macadamia tree (percentage difference of 2.67% between ALS- and TLS-derived results), because of its dense crown structure (Figure 4c).

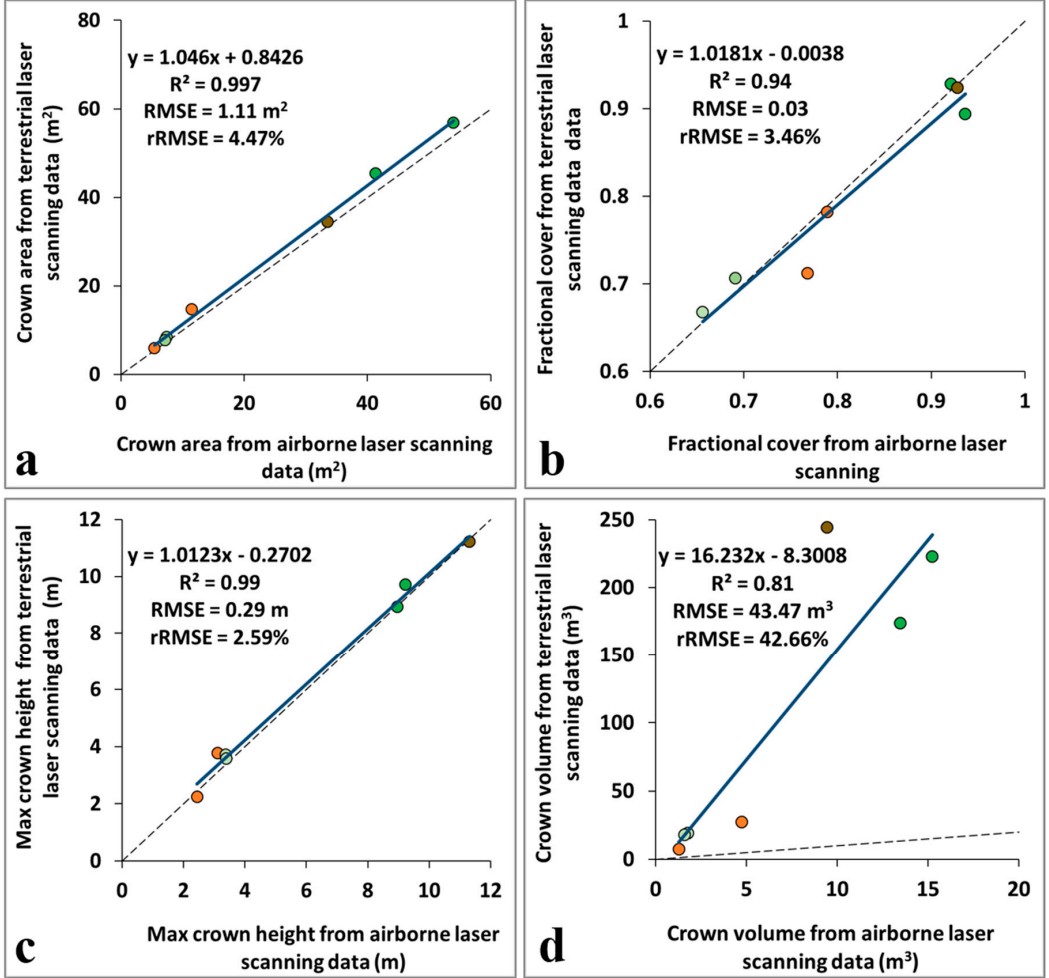

**Figure 3.** Crown area (**a**), fractional cover (**b**), maximum (max) crown height (**c**), and crown volume (**d**) compared between the airborne (ALS) and terrestrial laser scanning (TLS) data.

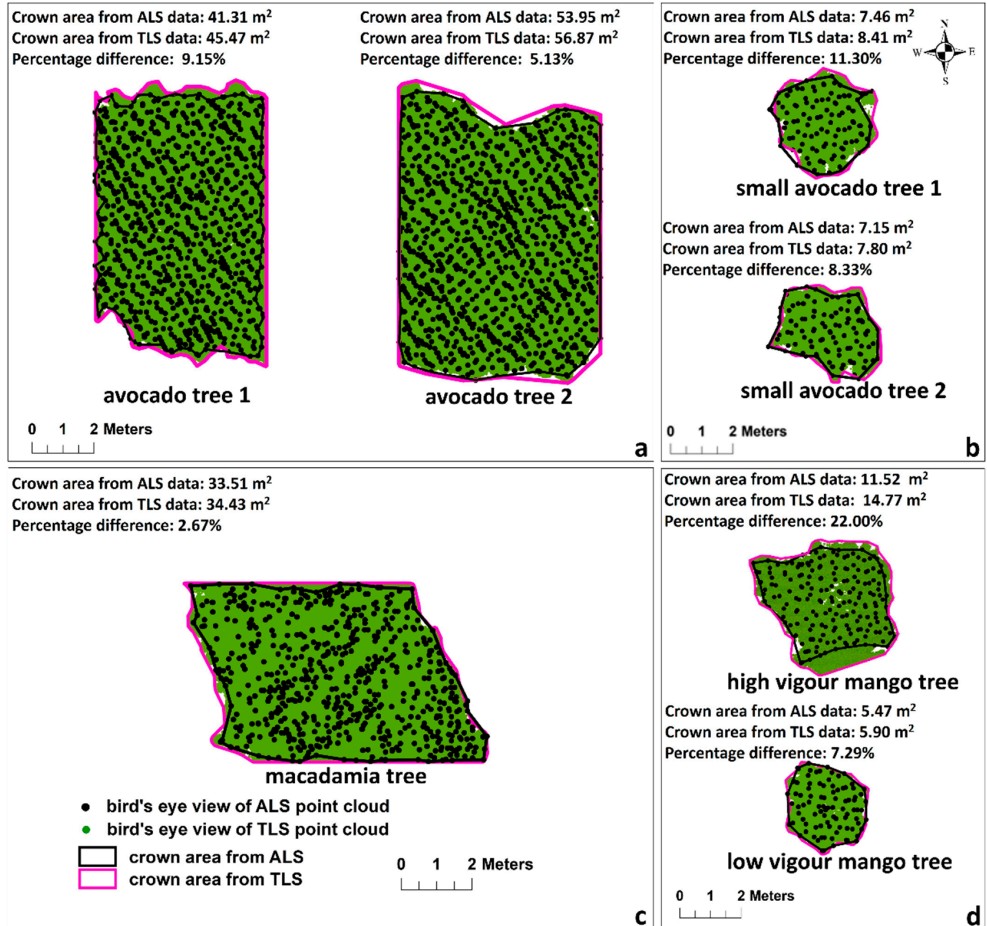

**Figure 4.** Crown area calculation of (**a**) the avocado trees from the commercial orchard and (**b**) the research station, (**c**) the macadamia tree, and (**d**) the mango trees with high and low vigour derived from the airborne (ALS) and terrestrial laser scanning (TLS) data.

### 3.2. Evaluation of ALS Fractional Cover Against TLS Data

Fractional cover measured from the ALS data was evaluated against the TLS data collected. At the tree crown level, there was no consistent underestimation or overestimation of fractional cover when relating the results between the ALS and TLS data (Figure 3b). Fractional cover along the crown boundaries had lower agreement between the results from the ALS and TLS data than those in the crown centre (Figure 5). This may be due to the lower point density of the ALS data, which may not hit the sparser crown perimeters, whereas the much denser point cloud of the TLS data increases the ability to map the exact crown perimeter. It seems that with the ALS data, it is more likely that the perimeter of sparse and small canopies is omitted, i.e., mango and small avocado trees, when compared with dense and tall canopies, i.e., mature avocado and macadamia trees (Figure 5). However, for all trees, differences of ≥0.84 in fractional cover were identified along the tree crown perimeters (Table 2). The perimeters of the tree crowns are also more likely to be influenced by wind, which may have caused the large maximum differences in fractional cover. In contrast, the crown centers displayed differences <0.10 between the ALS- and TLS-derived measurements. The average fractional cover of the pixels composing the individual tree crowns produced similar results for the ALS and TLS data, with the largest average difference between 0.04 for avocado tree 2 (Table 2). Besides the characteristics of the tree crown perimeters and size, ALS data still have a similar ability to the TLS data to estimate fractional cover at the tree crown level (Figure 3b), despite the larger RMSE between ALS- and TLS-derived measurements when assessing within-tree crown pixel values (Table 2).

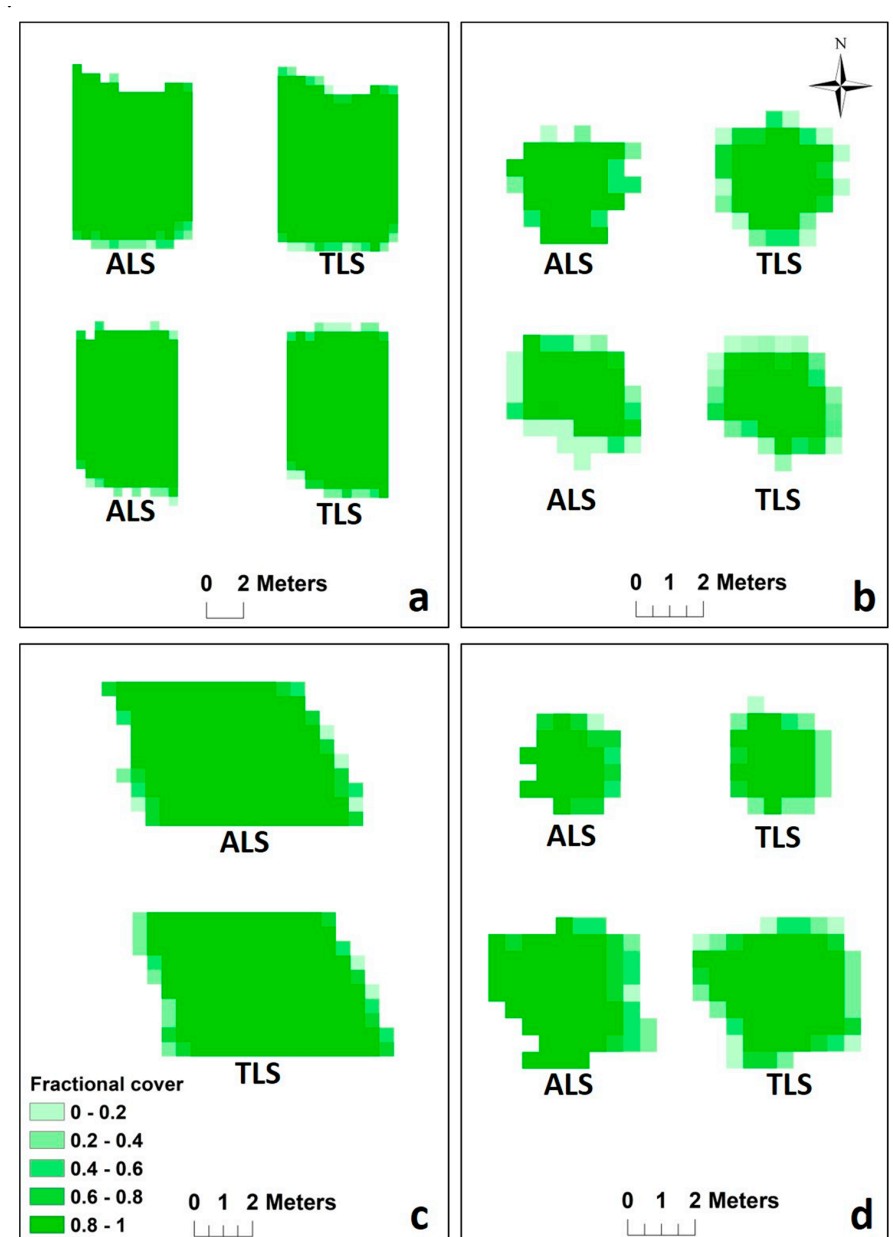

**Figure 5.** Fractional cover estimates of (**a**,**b**) avocado, (**c**) macadamia and (**d**) mango trees derived from airborne (ALS) and terrestrial laser scanning (TLS) data.

**Table 2.** Comparison of airborne- (ALS) and terrestrial laser scanning- (TLS) derived tree fractional (frac) cover at the individual pixel level for each of the avocado, macadamia and mango trees, showing the number of pixels per tree crown (n), maximum (max) and minimum (min) fractional cover difference between the ALS and TLS data, the average ALS and TLS fractional cover, and the root mean square error (RMSE) between the individual ALS and TLS pixel values per tree crown.

| | n | Max Frac Cover Difference | Min Frac Cover Difference | Average ALS Frac Cover | Average ALS Frac Cover | RMSE |
|---|---|---|---|---|---|---|
| Avocado tree 1 | 198 | 0.84 | 0 | 0.92 | 0.93 | 0.12 |
| Avocado tree 2 | 230 | 0.84 | 0 | 0.93 | 0.89 | 0.22 |
| Mango high vigour | 71 | 0.92 | 0 | 0.78 | 0.78 | 0.33 |
| Mango low vigour | 34 | 0.88 | 0 | 0.73 | 0.72 | 0.30 |
| Macadamia tree | 163 | 1 | 0 | 0.88 | 0.87 | 0.32 |
| Small avocado tree 1 | 40 | 1 | 0 | 0.69 | 0.71 | 0.34 |
| Small avocado tree 2 | 39 | 1 | 0 | 0.66 | 0.67 | 0.30 |

### 3.3. Evaluation of ALS Maximum Crown Height Against TLS Data

The maximum crown height of the avocado, macadamia and mango trees was calculated by extracting the value of the highest point within each tree crown for both the ALS and TLS data and subtracting the ground elevation from this height measurement (Figure 3c). The absolute height difference between these two datasets was 0.07 m–0.58 m. Some studies have found that ALS may underestimate crown height due to the lower point density and the larger foot print size in relation to crown structure [35]. To derive the maximum crown height from ALS data, a laser beam has to hit the tree apex and the return of this laser beam has to be recorded by the scanner [32]. Yu et al. [41] found that data acquisition parameters such as flying altitude and pulse density can affect the ability of height estimation from ALS data, and that canopy height underestimation increases with flying height. Our ALS- and TLS-derived height measurements of the avocado, macadamia and mango trees produced a near 1:1 relationship for maximum tree height (Figure 3c), which was attributed to the relatively flat semi-spherical crown top of the mango, mature avocado and macadamia trees. In some cases, TLS data can underestimate tall trees (e.g., 15 m [42]) due to possible occlusions of the upper crown [43]. However, when assessing the maximum values of individual pixels within each tree crown, there was a tendency of the TLS data producing larger height values than the ALS data, which was the main contributor to the RMSE reported in Table 3. In fact, all maximum differences (Table 3) in height values between the ALS- and TLS-derived measurements were due to height underestimation using the ALS data. Edge pixels showed the largest variation of height values with both over- and underestimation of ALS-derived measurements (Figure 6), which could possibly be attributed wind effects. For the pixels encompassing each tree crown, the TLS data produced larger height values in 86.96–97.50% of cases for six (not the low vigour mango tree) out of the seven assessed trees (Figure 6, Table 3). Because of these findings, and the fact that the horticultural trees were less than 12 m in height with sufficient space between tree rows for apex identification, TLS was found to be an appropriate technology for evaluation of ALS-derived crown height in this study. The larger TLS height values resulted in average heights per tree crown being between 0.39–0.94 m greater than those from the ALS-derived measurements, excluding the lower vigour mango tree (Table 3). These results highlight that while the maximum height value per tree crown produced a near 1:1 relationship between the ALS- and TLS-derived measurements; the within-crown ALS measurements were generally underestimated in relation to the TLS measurements.

**Table 3.** Comparison of airborne- (ALS) and terrestrial laser scanning- (TLS) derived tree height at the individual pixel level for each of the avocado, macadamia and mango trees, showing the number of pixels per tree crown (n), maximum (max) and minimum (min) height difference between the ALS and TLS data, the average ALS and TLS height, and the root mean square error (RMSE) between the individual ALS and TLS pixel values per tree crown.

| | n | Max Height Difference (m) | Min Height Difference (m) | Average ALS Height (m) | Average TLS Height (m) | RMSE (m) |
|---|---|---|---|---|---|---|
| Avocado tree 1 | 198 | 6.21 | 0.004 | 6.88 | 7.33 | 1.12 |
| Avocado tree 2 | 230 | 7.07 | 0.001 | 7.24 | 7.63 | 0.78 |
| Mango high vigour | 71 | 3.27 | 0.012 | 2.54 | 3.20 | 0.80 |
| Mango low vigour | 34 | 1.31 | 0.022 | 1.87 | 1.25 | 0.84 |
| Macadamia tree | 163 | 7.28 | 0.032 | 8.02 | 8.96 | 1.54 |
| Small avocado tree 1 | 40 | 1.64 | 0.040 | 2.47 | 2.93 | 0.59 |
| Small avocado tree 2 | 39 | 1.51 | 0.062 | 2.21 | 2.71 | 0.64 |

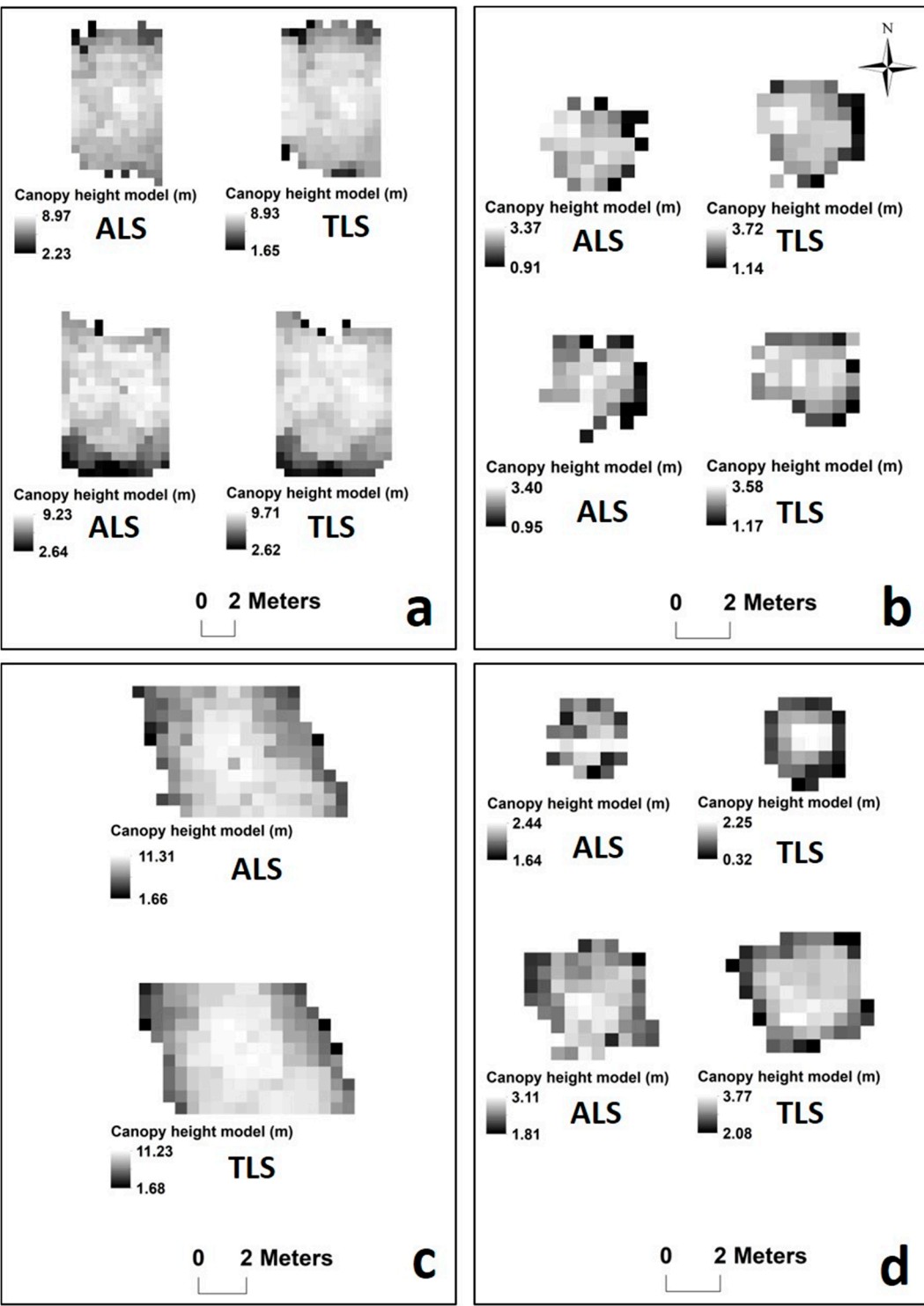

**Figure 6.** Crown height calculation of (**a**,**b**) avocado, (**c**) macadamia and (**d**) mango trees from airborne (ALS) and terrestrial laser scanning (TLS) data.

### 3.4. Evaluation of ALS Crown Volume Against TLS Data

The crown volume was calculated by counting the filled voxels (25 cm side length) for both the ALS and TLS data. This voxel size was considered appropriate in relation to the size of the assessed tree crops. The selection of an optimal voxel size to calculate crown volume of each type of tree crops is beyond the scope of this research, and therefore should be further investigated in subsequent studies.

The crown volume of the avocado, macadamia and mango trees derived from the ALS data was significantly smaller than that derived from the TLS measurements (Figure 3d). The underestimation increased for trees with overlapping canopies, high crown depth and density, e.g., within hedgerows such as the macadamia tree, where the crown volume calculated from the ALS data was only 3.86% of that measured from the TLS data. In contrast, the crown volume of the two mango trees derived from the ALS data was around 17% of that measured from the TLS data. For the two mango trees, we can see a general semi-spherical crown top from the ALS data, but the lower crown and individual leaves and branches cannot be identified (Figure 7). A similar semi-spherical crown top can also be seen for the mature avocado and macadamia trees, with only a limited number of ALS returns registered from within and the lower sections of the crown. The horizontal view angle, the multiple views, the small laser footprint and high point density of the TLS data produced a more evenly distributed point cloud across all parts of the crowns. Hence, crown volume based on the calculation of voxels could more appropriately be derived from the TLS than the ALS data. However, other methods for estimating crown volume may reduce the underestimation of the ALS-derived volumetric results. The crown sizes of the sampled orchards were either below 30 m$^3$ for the mango trees and the two 2-year-old avocado trees (Figure 7a,b) or between 150 m$^3$ and 250 m$^3$ for the mature avocado and macadamia trees (Figure 7a',b'). The associated pruning practices may have attributed to these distinct volumetric intervals.

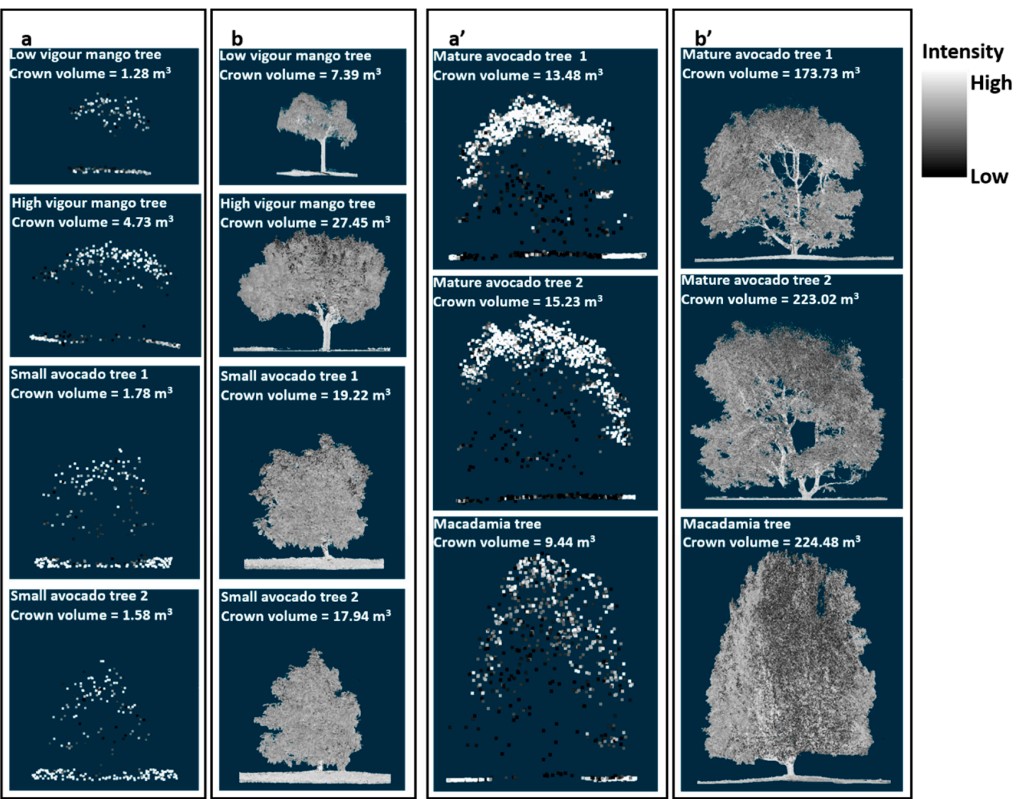

**Figure 7.** Crown volume (calculated at 25 cm voxel side length) and laser point clouds from airborne laser scanning data (**a**,**a'**) and terrestrial laser scanning data (**b**,**b'**) of two mango trees, two 2-year-old small avocado trees from a research station, two mature avocado trees from a commercial orchard, and one macadamia tree.

### 3.5. Additional ALS Orchard and TLS Canopy Parameters

The much larger spatial coverage of the ALS data enables orchard scale information to be derived. Figure 8 provides an example of how ALS data can be applied to obtain information on tree height variations as well as row widths and lengths within an orchard. Variations in tree height of individual

tree crowns can be seen along and between different rows, which might indicate differences in growth patterns and condition. The width of the rectangular bounding box, within which each tree row occurs, was determined by the widest tree crown in a row and can provide a quick overview of growth patterns and relative crown area variation between trees within a row. Tree length and width are useful parameters for informed pruning practices and orchard management. Based solely on the length of the rectangles forming each tree row and the known spacing between individual trees, a total of 495 mango trees, 4815 avocado trees and 3947 macadamia trees were estimated. All mango trees were correctly estimated based on manual counting of the mango trees within the CHM. While some smaller avocado trees were omitted along the perimeter of the orchard due to their limited height and size, a total of 4786 trees were estimated when deducting 29 trees based on observed gaps, i.e., missing trees, along the rectangles. Based on 561 field-counted avocado trees along 10 rows (rows 66–75, Figure 8b), a total of 559 were estimated from the delineated rows. Rounding the numbers of the rectangle length divided by the tree spacing, i.e., 5 m, was found to be the cause of the two unaccounted trees in this case. The estimated number of macadamia trees was 3899 when subtracting hedgerow gaps representing 48 missing trees within the orchard (Figure 8c). Based on 740 field counted macadamia trees along 5 rows (rows 18–23, Figure 8c), a total of 742 were estimated from the delineated rows. The overestimation of two trees based on the ALS data was due to two gaps, each representing two missing trees instead of one.

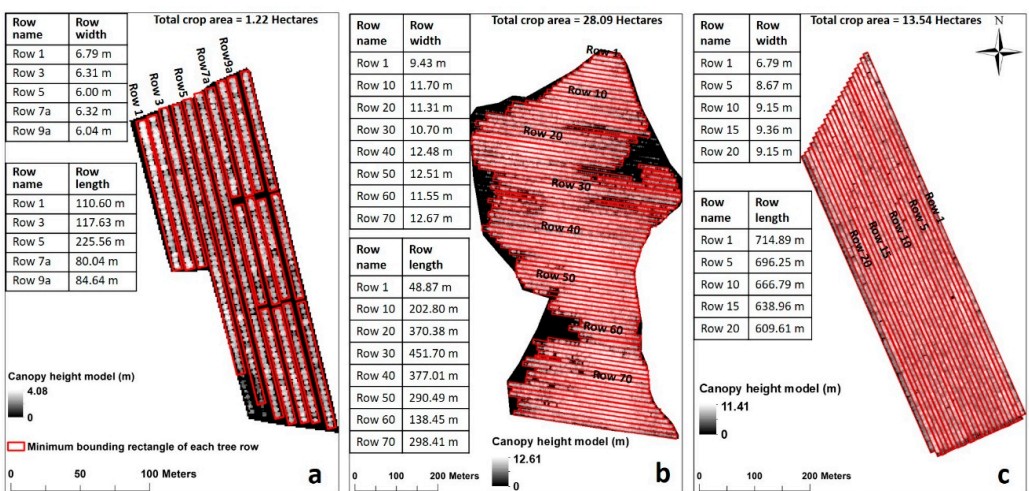

**Figure 8.** Canopy height and row width and length from airborne laser scanning of a mango orchard (**a**), an avocado orchard (**b**) and a macadamia orchard (**c**). Note that young trees in the mango and avocado orchards were excluded in this calculation.

Based on the methods and results by Wu et al. [6], some examples of additional information on tree structure that can be produced from TLS data is provided in Figure 9. The leaf area density was calculated for all parts of the tree crowns, displaying variations in leaf area in three dimensions. Relating leaf area calculations at the voxel level to tree height enables a vertical profile of leaf area to be produced, in this case showing that the densest parts of the crown with the highest leaf area occurred approximately 1 m above ground level for both of the 2-year-old avocado trees (Figure 9). However, this level of detail can only be obtained for selected trees, as covering an orchard with hundreds or thousands of trees would be prohibitively cost- and time-consuming, and the amount of data would be excessive. Hence, the ALS and TLS technologies provide different but complementary capabilities for mapping tree crop structural metrics to support orchard management at both individual tree crown scales and orchard scales.

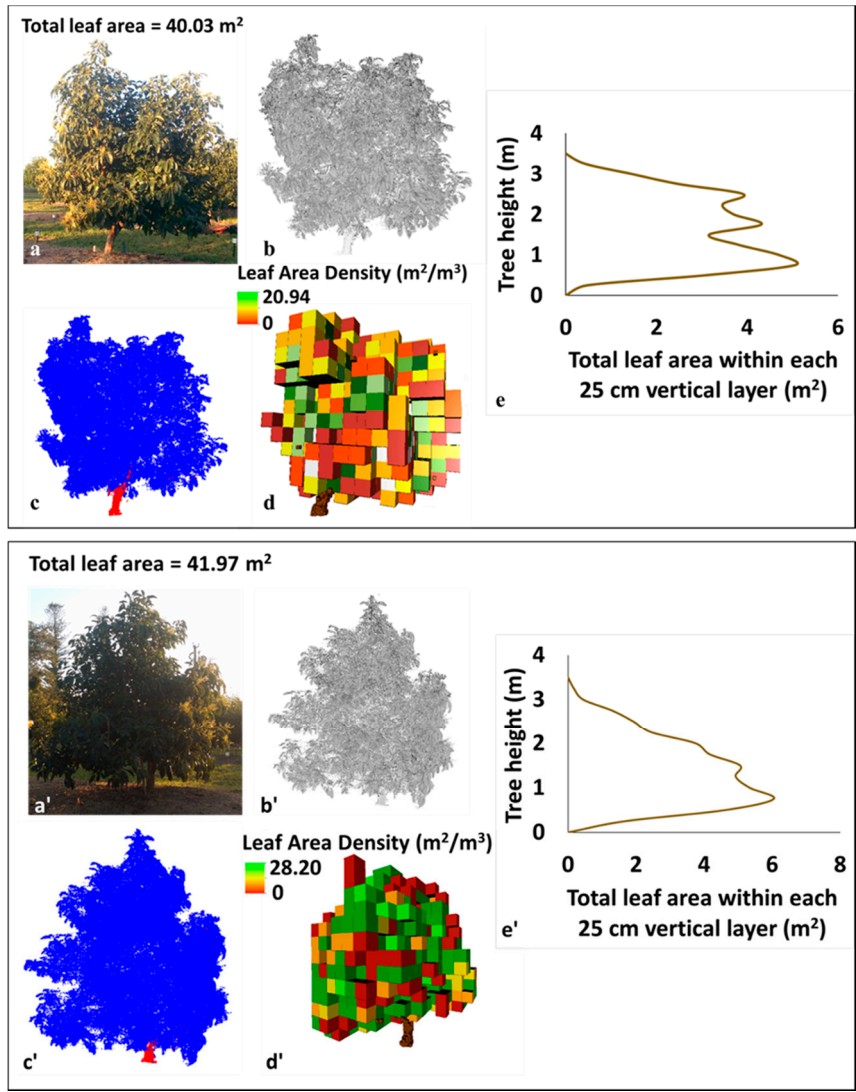

**Figure 9.** Pictures of two 2-year-old avocado trees (**a**,**a'**) from a research station, their corresponding 3D intensity images (**b**,**b'**) and classified discrete point clouds (**c**,**c'**), leaf area density at a voxel level (25 cm in side length) (**d**,**d'**), and vertical leaf area profiles within each 25 cm vertical layer (**e**,**e'**).

## 4. Discussion

### 4.1. Evaluating ALS-Derived Tree Crop Structure Against TLS Data

This study showed that accurate crown area measurements for horticultural tree crops can be derived from ALS data. These results concur with those previously reported for olive trees [18]. Crown area can be used as an indicator of tree growth during the growing season for horticultural tree crops, and hence provides information, if assessed on a multi-temporal basis, on tree response to orchard management practices, such as irrigation and fertilization [44]. Due to the irregular shape of horticultural trees, tree crown shape cannot simply be assumed to be circular or spherical [11]. Therefore, it is time intensive to manually conduct field measurements of crown area [45]. ALS data offer growers an accurate and efficient alternative. As the crown area calculation from the LiDAR data was mainly based on the upper crown returns, ALS, in this case with a point density of > 13 points/m$^2$, and TLS data demonstrated similar capacities to generate highly accurate crown estimates. However, TLS data collection is significantly more time-consuming and will often require a tree to be scanned from multiple angles to minimize occlusion issues [46]. Hence, mapping tree crown area from an aerial perspective seems more appropriate.

An alternative to our TLS data collection includes LiDAR-based mobile surveying technology consisting of a laser scanning system, generally with integrated positioning and imaging capabilities, and installed on a moving platform [47]. These types of systems have the capacity to collect point cloud data from multiple view angles to prevent occlusion and to significantly reduce the data collection time in orchards. While holding great potential for future point cloud data collection, the limited spacing between hedgerows restricts the type of vehicle or robotic system to be used for mobile mapping systems. For instance, Underwood et al. [48] used a TLS system mounted on a mobile robotic ground vehicle for almond orchards to calculate crown volume from a 3D voxelised point cloud from which yield was estimated. However, ground obstacles might hinder vehicle access or limit the quality of collected data [49]. While the objective of this research was to assess ALS data evaluated against TLS data for deriving structural parameters relevant for horticulture tree crops, it is important to highlight that crown area can be estimated from other types of image data, including airborne and UAV-based optical datasets [8,50–52] and potentially high spatial resolution satellite imagery [53]. However, Johansen et al. [54] found that high spatial resolution WorldView-3 pan-sharpened imagery with 30-cm pixels was not sufficient for delineating individual macadamia trees grown as a hedgerow.

The ALS dataset was found to provide highly accurate crown fractional cover estimates for horticultural trees when assessed against the TLS data (Figure 5, Table 2). LiDAR data have been widely applied in crown fractional cover mapping of forested and riparian environments due to its ability to extract spatially continuous vertical crown structure information even without field data [37,55–58]. Fractional cover is traditionally assessed from an aerial view, i.e., identifying the fraction of vegetation versus no vegetation from nadir-viewing, and hence ALS data are particularly suited for the derivation of this measurement. While TLS data were obtained with a much higher point density (>10,000 points/m$^2$) than the ALS data (13.63 points/m$^2$), multiple scans at different viewing angles are required to avoid occlusion and hence reduce the ability to calculate fractional cover on the side of the tree opposite to the scan location. Fractional cover is a good indicator of light interception, which is also a crucial determinant of crop growth and important for flowering, fruit maturation and quality as well as reducing disease and pest incursions for horticultural tree crops [59,60]. Remote sensing technologies used for fractional cover estimates of horticultural tree crops still mainly rely on passive optical sensors [59,61,62]. Challenges of optical imagery collected from different dates, including shadow effects, sensor viewing geometry, illumination angle differences and leaf phenology effects, do not exist for LiDAR data [37,58,63]. While optical imagery generally needs calibration and validation either from field data or ALS data for measuring fractional cover, ALS data can provide consistent results with similar accuracies to those derived from field measurements, e.g., using the LI-COR LAI-2000 Plant Canopy Analyzer or hemispherical photographs [37,56].

For maximum crown height measurements, similar results were produced using both the ALS and TLS data. Accurate crown height estimates have also been achieved from both ALS and TLS data for olive, walnut, mango and avocado trees, although some ALS-derived measurements can underestimate the maximum crown height [11,21,23,35]. Our results presented in Table 3 also showed that the within-crown ALS height measurements were underestimated in relation to the TLS measurements by up to 0.94 m on average per crown. Misclassified ground points (tall grass classified as ground), low point density and failure to identify the crown apex may be a cause for crown height underestimation of ALS data for horticultural tree crops [19,35]. Wang et al. [42] demonstrated that ALS data can provide accurate crown top height estimates regardless of tree height, crown shape and species. This corresponds with our findings that ALS data are suitable for estimating maximum crown height for horticultural trees of different age, canopy management regime, height, and tree types, in our case mango, avocado and macadamia trees. Maximum crown height and width are controlled based on row space in commercial orchards to achieve the best light interception, machine access and convenience for harvesting [64–66]. Therefore, in mango orchards most trees are limited to a maximum height of 4 m, whereas avocado and macadamia trees might grow to around 10 m. While the footprint size of ALS returns, the pulse repetition rate, flying height and the threshold beyond which a return is registered

will affect the ability of ALS data to identify the highest point of tree crops, these parameters will rarely impact TLS data [67]. However, depending on the tree height and view angle of the scanner, the line of sight of TLS data might in some cases be restricted from identifying the apex of tall tree crops [38]. Alternative options for measuring tree height from remote sensing include the use of multi-view digital photography, in particularly structure-from-motion algorithms based on optical UAV imagery, where photogrammetry permits height measurements of tree crops to be calculated [45,50,68,69].

Laser scanning data are perfectly designed for obtaining point clouds for provision of 3D tree structure information such as crown volume. However, our results showed that the ability to measure crown volume is significantly affected by the point density and the ability of points to penetrate the crown. Although an ALS pulse has the capacity to reach the ground through open or sparse canopies, the bottom and inner crown parts were in most cases not detected by the ALS data, as the laser pulses were mainly returned by the upper parts of the crown [67]. Due to the occlusion issue and the insufficient point density, the ALS data used in this research could not detect the lower crown parts and hence underestimated crown volume using the voxel-based approach (Figure 7). Similar findings have also been identified in ALS and TLS comparison studies for calculating canopy volume in forestry [70]. By demonstrating the different viewing and data capture geometry from ALS and TLS data, Kükenbrink et al. [70] concluded that occlusion effects were the main reason for canopy volume underestimation from ALS data. Goodwin et al. [67] indicated that higher canopy density and canopy cover reduce beam penetration through the canopy and thus reduce the point density from the bottom canopy. Therefore, the macadamia tree, which had the highest crown density and the highest crown depth of the trees assessed in this study, showed the largest relative difference in crown volume between the ALS and TLS data (Figure 7). As crown volume is directly related to tree health and vigour [71], our crown volume results of the high and low vigour mango trees reflect this point as well, where the high vigour mango tree was 3.7 times larger than the low vigour mango tree (Figure 7). In addition, voxel size is a critical parameter for voxel-based crown structure measurements [6,16,20]. Future work should include further experiments on the optimal voxel size selection based on crown size, branch structure, occlusion effect and scanning resolution. Colaço et al. [72] tested different canopy volume calculation methods (convex-hull and the alpha-shape surface reconstruction algorithms) for orange trees and found substantial volume differences based on different methods. Therefore, more canopy volume modelling methods need to be further evaluated in the future to determine which ones are most accurate and how they are affected by different crown structure and size.

Differences in wind direction and speed between the days that the ALS and TLS data were collected could have caused differences amongst the mapped tree crown perimeters due to movement of branches and leaves [73]. The wind speed measured from a nearby climate station (Bundaberg Aero weather station; 24.9069°S, 152.3230°E) when scanning the two avocado trees in the commercial orchard, two small avocado trees from the research station, one macadamia tree and two mango trees were 20.5–38.9 km/h, 24.1–27.7 km/h, 9.4–13 km/h and 29.5–40.7 km/h, respectively. The average wind speed from the same weather station at the time of the ALS data collection was 11.8 km/h. However, the hedgerow structure of the orchards acts as a barrier for blocking the wind. Although the wind speed was higher during the TLS scanning, the two avocado and the macadamia trees (Figure 6) were well sheltered amongst other tree rows, which likely contributed to the good agreement between the ALS- and TLS-derived tree crown heights. The two small avocado trees from the research station and the two mango trees may have been more influenced by wind, as they were more exposed. This might have contributed to the offsets been crown apex and perimeter observations between the ALS and TLS data (Tables 2 and 3).

### 4.2. Capacity of ALS and TLS Data for Imporved Orchard Management

TLS data can bridge the gap between traditional field measurements and ALS data [38], and the combined application of ALS and TLS data provides an alternative method for accurate 3D characterization of horticultural tree crops over both local and orchard scales. It is time-consuming,

inconsistent and impractical to manually measure horticultural crown structure in situ over a large area [38]. Our results showed that TLS data with high-resolution point clouds can be used for evaluation of tree crown structure measurements derived from ALS data [39], while ALS data can upscale these highly accurate measurements from crown to orchard scales (e.g., Figure 8 showing canopy height of tree rows). Row width and canopy height are essential parameters that can be used to guide horticultural tree pruning [64–66]. Having measurements of row width and length can provide growers with an estimate of the total tree crop area and number of trees as shown in our results, which in other studies have proven useful for yield estimation, yield forecasting and sprayer calibration [46,74]. On the other hand, ALS data cannot provide detailed 3D crown structure information, such as voxel level LAD and leaf area profile mapping, whereas the much denser point cloud (>10,000 points/m$^2$) derived from TLS data is ideally suited for these tasks [6,75] (Figure 9). Detailed crown information is useful for targeted limb removal, precision pruning for improved light interception, irrigation, fertilization and pesticide applications [4,6]. As such, while TLS data can be used to calibrate and validate ALS-derived information, our results clearly show that both the ALS and TLS technologies have potentialities for joint use to improve orchard management at different spatial scales. Further research should expand on our work to implement our research findings in an operational manner to evaluate the cost-benefits of the joint use of ALS and TLS data for orchard management and capacity to assess and improve crop production.

The accuracy of canopy structure mapping from ALS data is often based on the data density for horticultural trees, shrubs and forestry [10,67,75,76]. Estornell et al. [18] reported that compared to forest structure mapping, which focuses on plots, higher point density may be required to predict crown structure information of individual horticultural tree crops and that a point density of 4 points/m$^2$ had the capacity to estimate crown area and pruning biomass. Hadaś et al. [10] assessed different ALS point densities (0.5, 3.5 and 9 points/m$^2$) for mapping tree height, crown base height, average crown diameter, and crown area of olive trees and concluded that ALS data with a higher point density are better suited for measuring crown structure, especially crown height and base height. With a point density of 13.63 points/m$^2$, our results showed that top of tree crown parameters such as crown area, fractional cover and height can be accurately mapped. However, these structural parameters could not be mapped in an automated manner for individual tree crowns for the avocado and macadamia trees because of their hedgerow structure with adjoining tree crowns preventing automated crown delineation. Other studies using UAV-based optical data and structure-from-motion generated point clouds have also reported similar difficulties with regards to individual tree crown delineation for macadamia, olive and avocado hedgerows [54,68,77]. While ALS data with higher point cloud densities and small laser footprints may improve the ability to delineate individual tree crowns within orchards with hedgerow, orchards planted at different densities, and canopies with different structures and sizes should be studied separately, as these are factors that affect the ability of ALS data to capture lower canopy layers [78]. Further experiments on point density requirements of different tree crops in relation to planting densities should be undertaken in future studies.

While our results demonstrated the capability of ALS to produce orchard scale maps of tree structure, unmanned aerial vehicles (UAVs) provide another remote sensing platform suited for integration with miniaturized sensor systems that offer potential opportunities for frequent and more cost-effective monitoring of orchards [10,11,75]. Optical based structure-from-motion information acquired from UAVs have also demonstrated the capability for accurate crown structure mapping in orchards, including parameters such as crown area, tree height and crown volume of horticultural trees for orchard scale applications [45,50,77]. However, UAV-based optical data and derived point clouds and digital surface models are more likely to prove beneficial for top of canopy information extraction. Studies have shown that UAV-based LiDAR data with high point density (3200 points/m$^2$) can successfully provide crown identification (92% of trees) and highly accurate crown height estimates (RMSE = 0.09 m, R$^2$ = 0.96) for horticultural tree crops [79]. Therefore, UAV-based data may provide an alternative to ALS and TLS data for orchard scale (<1 km$^2$) canopy structure mapping of horticultural

tree crops. Future research should compare UAV-based LiDAR mapping accuracy of tree crop structure with those derivable from ALS and TLS data as shown in our research. Hadas et al. [80] used a UAV-based LiDAR system to successfully identify 99% of 655 apple trees, and then selected 50 trees for mapping crown structure, including crown area, tree height and crown base height, with high accuracies. They also found that UAV-based LiDAR data tend to underestimate crown area when compared against field measurements (RMSE = 1 m$^2$, R$^2$ = 0.17), while tree height (R$^2$ = 0.96, RMSE = 0.09 m) and crown base height measurements achieved better accuracies [80]. As the cost of UAV-based LiDAR systems drops in the future, it is likely they will become operational for commonplace tree crown structure assessment. For UAV-based LiDAR mapping of crown structure, TLS technology will likely provide a suitable means for such applications in the future to ensure high quality calibration and validation data.

## 5. Conclusions

This study demonstrated and assessed the ability of ALS data evaluated against TLS data for mapping crown structure metrics (crown area, fractional cover, crown height, and crown volume) of individual horticultural tree crops, including avocado, macadamia and mango trees. In the evaluation of the ALS-derived results against the TLS measurements, we found significant agreement between estimates of crown area, fractional cover and maximum crown height. However, the use of ALS data significantly underestimated crown volumes of horticultural tree crops when evaluated against TLS data, especially for the macadamia tree, which exhibited the highest crown density. The ALS data were found suitable for measuring horticultural crown structural parameters mainly relying on top crown information as well as hedgerow width, length and number of trees at the orchard scale. In contrast, TLS data did not have the capacity to map crown structure over a large area but were found suitable for assessment of all crown strata, which is required to measure crown volume, LAD and vertical leaf area profile of individual trees. It is suggested that TLS data may replace traditional field measurements for calibration and validation of ALS-derived crown structure measurements and applied jointly with ALS data for individual tree scale assessment of structural parameters. One limitation of this study was that only seven trees for the crown structure measurements were measured with TLS, which was due to the time-consuming exercise of collecting high-resolution TLS data from multiple scan angles. However, the tree types and orchards we chose for this study represented different canopy structure complexity, planting density and crown size, demonstrating the suitability of TLS data for deriving structural measurements for tree crops with a large variety of planting regimes and structural complexities. Future experiments should be based on larger sample sizes and might compare additional crown structure information such as LA and leaf area index as well as optimization of ALS and TLS acquisition settings and structural parameter information extraction methods. Finally, further research should explore UAV-based platforms for acquisition of laser scanning data suited for management of tree crop orchards.

**Author Contributions:** Conceptualization, D.W., A.R., S.P. and K.J.; methodology, D.W., K.J. and S.P.; software, D.W.; validation, D.W.; formal analysis, D.W., K.J., S.P. and A.R.; investigation, D.W.; resources, D.W., A.R., S.P. and K.J.; data curation, D.W.; writing—original draft preparation, D.W., K.J.; writing—review and editing, K.J., A.R., S.P.; visualization, D.W.; supervision, S.P., K.J., A.R.; project administration, S.P., A.R.; funding acquisition, A.R., S.P. All authors have read and agreed to the published version of the manuscript.

**Funding:** This research was funded by Department of Agriculture and Water Resources, Australian Government as part of its Rural R&D for Profit Program's subproject, titled "Multi-Scale Monitoring Tools for Managing Australian Tree Crops–Industry Meets Innovation", grant number RnD4Profit-14-01-008.

**Acknowledgments:** The authors acknowledge the Australian Federal Government "Rural R and D for Profit" scheme and Horticulture Innovation Australia for funding this research. The authors appreciate the support, especially during field trips, provided for this research by Chris Searle from MacAvo Consulting, by Simpson Farms Pty. Ltd. (Childers, QLD 4660, Australia), in particular, Chad Simpson, Bundaberg Research Facility, and by the Queensland Government's Department of Agriculture and Fisheries, in particular John Wilkie and Helen Hofman. We thank Martin Béland for sharing the MATLAB code to calculate the leaf area density, Peter Scarth for

MATLAB assistance and Nicholas Goodwin for assisting with data registration. We also thank Aaron Aeberli and Yu-Hsuan Tu for their assistance with fieldwork.

**Conflicts of Interest:** The authors declare no conflict of interest. The funders had no role in the design of the study; in the collection, analyses, or interpretation of data; in the writing of the manuscript, or in the decision to publish the results.

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
