# Peer review of "Suitability of Airborne and Terrestrial Laser Scanning for Mapping Tree Crop Structural Metrics for Improved Orchard Management"

_remotesensing, doi:10.3390/rs12101647_

Round 1

Reviewer 1 Report

The manuscript under review focuses on an interesting and really essential topic, as is the potential capacity of using LIDAR methods (either airbone or terrestrial methods) to assessing structural parameters on horticultural tree crops. In this sense, the general topic of the text is worthwhile and falls within the scope of Remote Sensing journal. Despite this, the reality after evaluating the manuscript is that the scope of the work is much narrower than what one could expect from the title. The manuscript merely presents a comparison of four interest parameters (crown area, fractional cover, canopy height and canopy volume) derived from ALS and TLS measurements over uniquely seven trees from three different species, which seems quite a short number for these types of analysis. Results from the analysis are somewhat expected and logical: ALS limitations for measuring crown volume. These results are accompanied by a huge and exhaustive graphical description of the data from each tree (figures 4-7), with no statistical analysis at all. On the other hand, very little information is derived from the results obtained focusing on the possible application and upscaling of these techniques to improve the practical management of orchards. In this sense, I think that while the techniques and methodology presented are essentially correct and correctly applied, the work lacks of a more innovative and ambitious approach.

In the next paragraphs I’ll expose my main concerns with respect to the manuscript, as well as some ideas or suggestions for improving the work.

MAJOR COMMENTS

Introduction, general Aims and proposed approach

When reading the manuscript it seems as if the main aim of the work is not to compare TLS and ALS techniques, but show the potentiality of ALS techniques in assessing structural parameters for crop trees, and evaluate this capacity using 7 trees where TLS measures have been recorded. So why not to orientate the manuscript in this sense? Why focus all the work on the comparison among ALS and TLS in those 7 trees, and ignore topics as the ability of ALS for determining so relevant parameters for a tree crop orchard as number of trees, crown width, canopy height, accuracy of crown delineation and tree individualization, etc… but referred to the whole plantation?. In this sense I strongly miss the comparison among ALS data and field data concerning these parameters, and analysis that will help to evaluate the potentiality of ALS something that is presented in section 3.5, but not developed at all.  

In addition I don´t agree with the idea of orientating the manuscript as a combat between both techniques, when it is evident that each technique has its own advantages and limitations. In this sense I think is much more interesting focuses on the potentiality of the joint use of both techniques to improve crop management, something mentioned in the text but not developed.

A suggestion: why not to analyse (or at least mention) the potentiality of these TLS or ALS methods to improve the assessment or predictions of crop production? (see e.g. Schneider et al. 2020)

No sound hypotheses are presented, something that is relevant in a scientific work. What do the authors expect to obtain from their analysis?

Material and methods

As previously indicated, seven trees from three different species seems a quite small sample as for carrying out relevant inference. The authors should justify in a deeper detail why they have used quite a low number of trees. Of course TLS measurements and (especially) data processing are large-time consuming, but why not to propose alternatives as e.g. reducing point density, or others?

Even despite the shortage of data, the authors could have entered in a deeper analysis at within crown scale. Why uniquely compare tree fractional cover and maximum height (a single record per tree), instead of carrying out comparison among the TLS and ALS observed distribution of frequencies of crown heights or fractional cover (information included in figures 5 & 6, and not statistically analysed)?

Lines 165-170: Relevant information on ALS concerning point density is missed.

Lines 174-177: It’s not clear the accuracy in matching ALS and TLS data at tree level. Which is the capacity of ALS in detecting the subject trees? Can be the authors completely confident in having matched the trees from both approaches? Related with the previous, no field data measurements are presented as a control of ALS and TLS data. The authors should justify in detail this decission.

Lines 229-230: mean error (mean value of observed TLS minus observed ALS) as well as level of significance associated will be a real useful statistics for testing bias on the methods

Results

Line 244: I strongly recommend the authors not to carry a simple visual assessment of the line 1:1, but a more detailed analysis studying if in the linear regression TLS = a + b ALS parameter “a” is not significantly different from zero and parameter “b” is not significantly different from 1 (separate t-tests of simultaneous F-test, see Yang et al. 2004 for further details)

Sections 3.1 – 3.4 are largely descriptive. Please reduce the length of the text (figures are enough for explaining)

Lines 284-285: is there any statistical analysis at within-crown scale? (see my comments above)

Line 295: The authors are analysing maximum tree crown height, not canopy height, what refers to the height of a stand, or a group of trees

Lines 265-272 , 300-306: Move to discussion

Section 3.5: this section refers to results not presented nether in material nor in methods section. In addition, the section uniquely shows some potentialities or abilities of the TLS technique, but the results are not contrasted with real field data. Reduce and move to discussion (or follow my suggestion above of including field data to contrast)

Discussion

While some discussion is spread through the results section (see e.g. lines 300-306) the disuccion itself is not a real one. The results from the analysis are not discussed against previous literature, but basically the section is an in-depth review of the current state of the art on TLS and ALS methods applied to horticultural crops, what has its own place on the introduction.

Lines 453-467 are largely speculative. Please reduce or remove

Section 4.2. is not at all a discussion of the obtained results, but a mere enumeration of potentialities of ALS and TLS, derived from previous scientific literature. In this sense, it seems as if the results from the current work do not participate in the “improvement of crop orchard management”

MINOR COMMENTS

Line 49: “crown diameter” instead of “diameter”

Line 87: please define or give a reference for Plant Area Index

Line 100: leaf area

Figure 2: Create canopy height model (> 0.50 m height…) (according to text)

Figure 4. Show figure 4C in the same scale than the others

Line 428: you don´t compare point density, so you cannot infer that canopy volume is significantly affected by point density

Line 443-445: This reference to tree health and vigour is out of place here        

REFERENCES

Schneider R. et al. 2020. Understanding Tree-to-Tree Variations in Stone Pine (Pinus pinea L.) Cone Production Using Terrestrial Laser Scanner. Remote Sensing 12: 173

Yang Y et al. 2004. An evaluation of diagnostic tests and their roles in validating forest biometric models Can. J. For. Res. 34: 619–629 (2004)

Author Response

Please find attached our responses to the comments made by Reviewer 1

Reviewer 2 Report

The study is comparing TLS and ALS data for retrieving tree crop structural characteristics for Orchard Management. For most of the characteristics derived in this study both of the data are suitable. The advantage of ALS data is in ability of analyze large area in short time.

The study is missing any field data validating the comparison of TLS and ALS data. There is also probably wrongly used method for computation of canopy volume in case of ALS data, which caused significant underestimation of canopy volume based on ALS data. 

Here I listed some notes and recommendations for improvement of the paper:

line 43: typo: import-> important ?

line 48: Is it correct "canopy detection"? It is more often in forestry to detect individual trees

line 99: it also takes more time to acquire more dense TLS point cloud data

line 122: abbreviations LAD and LA was already defined

line 126: It would be much better if the validation be realized on the other set of the data. At least most of the characteristics in this study are able to be measured by different than LiDAR technology.

line 161: Which validation? You mentioned here that seven trees were selected for validation purposes, but how they were measured, how the validation was done etc. There is no further mention of this validation.

line 165-170: Is it possible to add the information about point density per m2?

line 178-179: How precise is the segmentation?

line 183-184: From the mentioned references is still not clear how do you calculate the LA. You should cite directly publication of Béland et al. 2011. There is cited Wu et al. 2018, then Béland et al. 2014 and after that Béland et al. 2011, where are the exact explanation and validation of the method.

  • Wu, D., Phinn, S., Johansen, K., Robson, A., Muir, J., & Searle, C. (2018). Estimating Changes in Leaf Area, Leaf Area Density, and Vertical Leaf Area Profile for Mango, Avocado, and Macadamia Tree Crowns Using Terrestrial Laser Scanning. Remote Sensing, 10(11), 1750. https://doi.org/10.3390/rs10111750
  • Béland, M., Widlowski, J.-L., & Fournier, R. A. (2014). A model for deriving voxel-level tree leaf area density estimates from ground-based LiDAR. Environmental Modelling & Software, 51, 184–189. https://doi.org/10.1016/j.envsoft.2013.09.034
  • Béland, M., Widlowski, J.-L., Fournier, R. A., Côté, J.-F., & Verstraete, M. M. (2011). Estimating leaf area distribution in savanna trees from terrestrial LiDAR measurements. Agricultural and Forest Meteorology, 151(9), 1252–1266. https://doi.org/10.1016/j.agrformet.2011.05.004

Section 3.1. Comparison of Crown Area from Airborne and Terrestrial Laser Scanning Data: Is it possible to compare the differences between ALS and TLS data on some relative value? Compared trees are not all in the same size, the difference between the smallest and largest ones is significant. It would be better to describe the difference between TLS and ALS based crown area, for example, as percentage from TLS.

line 272: the reference to the Figure 4 could be specific as Figure 4d or 4D as it is mentioned only high vigour mango tree

Figure 4A: The second avocado tree (on the right) is also marked as "avocado tree 1" it would be probably "avocado tree 2", right?

Figure 4 (and the other Figures): On previous Figures 1 and 3 the segments of the Figure are marked with small letters (a,b,c,d), why it is changed to capital letters (A,B,C, D) on Figure 4? It is not consistent through all figures. Even later there are new one using A,B,A', and B'.

Section 3.2. Comparison of fractional cover ...: There are no numerical evaluation of the comparison (except the one for whole tree on Figure 3b). Is it possible to calculate some pixel statistics and compare them?

Line 299: The difference between the two data sets should be described in relative values not absolute, if there are significant differences in size of the compared trees.

line 331-333: There is probably much significant influence of chosen method and voxel size than the differences between ALS and TLS data. Therefore the sentence at these lines could be rephrased. Or at least mention, that there could be also other influences and the ALS data used with different method could be also sufficient or at least not that much underestimated.

line 376: "..., ALS (with reasonable point density) ..." Could you mention some numbers or range?

line 434: Is it possible, when the larger voxels were used, then the canopy volume would not be that underestimated in case of ALS data? Did you consider it? In the next sentence you referred publication, where they conclude the same as you. Nevertheless they perform sensitivity study of voxel size and they choose optimal voxel size 1m3. Also their research is focused on much older, higher trees.

References:
[37] Yu, X.; Hyyppä, J.; Hyyppä, H.; Maltamo, M. Effects of flight altitude on tree height estimation using airborne laser scanning; 2012; Vol. 36.

  • The published year is 2004 according to the link  https://www.isprs.org/proceedings/XXXVI/8-W2/ 
  • there are missing information where it is published "ISPRS - International Archives of the Photogrammetry, Remote Sensing and Spatial Information Sciences"
  • the page numbering "96-101"

[39] Krooks, A.; Kaasalainen, S.; Kankare, V.; Joensuu, M.; Raumonen, P.; Kaasalainen, M. Predicting tree structure from tree height using terrestrial laser scanning and quantitative structure models; 2014; Vol. 48.

  • there are missing information according to:
  • Krooks, A., Kaasalainen, S., Kankare, V., Joensuu, M., Raumonen, P., & Kaasalainen, M. (2014). Tree structure vs. Height from terrestrial laser scanning and quantitative structure models. Silva Fennica, 48(2). https://doi.org/10.14214/sf.1125 (it is in different format than yours)

[40] Lordan, J.; Pascual, M.; Fonseca, F.; Montilla, V.; Papió, J.; Rufat, J.; Villar, J.M. An Image-based Method to Study the Fruit Tree Canopy and the Pruning Biomass Production in a Peach Orchard. 2015, 50, 1809,
doi:10.21273/hortsci.50.12.1809.

  • missing pages range and journal:
  • Lordan, J., Pascual, M., Fonseca, F., Montilla, V., Papió, J., Rufat, J., & Villar, J. M. (2015). An Image-based Method to Study the Fruit Tree Canopy and the Pruning Biomass Production in a Peach Orchard. HortScience, 50(12), 1809–1817. https://doi.org/10.21273/HORTSCI.50.12.1809

[52] Trout, T.J.; Johnson, L.F.; Gartung, J. Remote Sensing of Canopy Cover in Horticultural Crops. 2008, 43, 333, doi:10.21273/hortsci.43.2.333.

  • missing pages range and journal:
  • Trout, T. J., Johnson, L. F., & Gartung, J. (2008). Remote Sensing of Canopy Cover in Horticultural Crops. HortScience, 43(2), 333–337. https://doi.org/10.21273/HORTSCI.43.2.333

[58] L. Davenport, T. Pruning Strategies to Maximize Tropical Mango Production From the Time of Planting to Restoration of Old Orchards; 2006; Vol. 41, pp. 544-548.

  • missing journal
  • Davenport, T. L. (2006). Pruning Strategies to Maximize Tropical Mango Production From the Time of Planting to Restoration of Old Orchards. HortScience, 41(3), 544–548. https://doi.org/10.21273/HORTSCI.41.3.544

[70] Rosca, S.; Suomalainen, J.; Bartholomeus, H.; Herold, M. Comparing terrestrial laser scanning and unmanned aerial vehicle structure from motion to assess top of canopy structure in tropical forests; 2018; Vol. 8, pp. 20170038.

  • missing journal
  • RoÅŸca, S., Suomalainen, J., Bartholomeus, H., & Herold, M. (2018). Comparing terrestrial laser scanning and unmanned aerial vehicle structure from motion to assess top of canopy structure in tropical forests. Interface Focus, 8(2), 20170038. https://doi.org/10.1098/rsfs.2017.0038

Author Response

Please find attached our responses to the comments made by Reviewer 2

Reviewer 3 Report

The manuscript describes the suitability assessment of LiDAR point clouds against TLS for horticulture trees management. The article is of interest, but there are two issues to be clarified:

  • Authors used and discussed the employement of TLS to derive canopy structure information, but they did not comment the use of portable mobile mapping systems. Since horticultural trees have short stems (<12 m in the present article), these systems could be an interesting solution to reduce the time-consuming field acquisition (line 378), improve the completness (line 394), and increase the numebr of trees mapped (line 32). Authors should discuss, or at least comment, this alternative solution.
  • I failed to find the precision of the targets employed to register the TLS point cloud to the same Coordinate Reference System that the LiDAR (I also did not find it in ref [6]). Please state the precision of the TLS-Lidar registration and assess if it has effect in the derived parameters. Similarly, authors stated the use of “multi Station Adjustment tools”, but they should expand it, indicating what kind of transformation was employed (six parameters? seven?).

Author Response

Please find attached our responses to the comments made by Reviewer 3

Round 2

Reviewer 1 Report

This is my second review of this interesting mansucript and I must to indicate that the current version has been substantially improved, so I do not detect any concern at all preventing its publication. I´d like to remark the effort made by the authors in improving the text (including new analysis as those presented in tables 2 & 3, or the capacities of ALS methods in detecting number of trees). In this sense I want to thank the authors for having considered the majority of the suggestions I made in the previous reviewing report, or having thoroughly justified those topics of discussion where we were in disagreement.

My felicitations for this nice work